# A New Stilbene Glucoside from Biotransformation-Guided Purification of Chinese Herb Ha-Soo-Oh

**DOI:** 10.3390/plants11172286

**Published:** 2022-09-01

**Authors:** Jiumn-Yih Wu, Hsiou-Yu Ding, Tzi-Yuan Wang, Min-Hui Hsu, Te-Sheng Chang

**Affiliations:** 1Department of Food Science, National Quemoy University, Kinmen County 892, Taiwan; 2Department of Cosmetic Science, Chia Nan University of Pharmacy and Science, No. 60 Erh-Jen Rd., Sec. 1, Tainan 71710, Taiwan; 3Biodiversity Research Center, Academia Sinica, Taipei 11529, Taiwan; 4Department of Biological Sciences and Technology, National University of Tainan, Tainan 70005, Taiwan

**Keywords:** *Polygonum multiflorum*, Ha-Soo-Oh, tyrosinase, biotransformation, hydroxylation, melanogenesis, antioxidant

## Abstract

Ha-Soo-Oh is a traditional Chinese medicine prepared from the roots of *Polygonum multiflorum* Thunb. The herb extract has been widely used in Asian countries as a tonic agent and nutritional supplement for centuries. To identify new bioactive compounds in Chinese herbs, the biotransformation-guided purification (BGP) process was applied to Ha-Soo-Oh with *Bacillus megaterium* tyrosinase (*Bm*TYR) as a biocatalyst. The result showed that a major biotransformed compound could be purified using the BGP process with preparative high-performance liquid chromatography (HPLC), and it was confirmed as a new compound, 2,3,5,3′,4′-pentahydroxystilbene-2-*O*-*β*-glucoside (PSG) following mass and nucleic magnetic resonance (NMR) spectral analyses. PSG was further confirmed as a biotransformation product from 2,3,5,4′-tetrahydroxystilbene-2-*O*-*β*-glucoside (TSG) by *Bm*TYR. The new PSG exhibited 4.7-fold higher 1,1-diphenyl-2-picrylhydrazine (DPPH) free radical scavenging activity than that of TSG. The present study highlights the potential usage of BGP in herbs to discover new bioactive compounds in the future.

## 1. Introduction

Isolating new compounds is the first step in the development of new drugs from herb plants. Traditionally, there are two strategies to isolate new compounds: searching via natural compounds and chemical synthesis in a laboratory. In addition to the above-mentioned methods, biotransformation is another strategy to create new compounds [1,2,3]. Through biotransformation, known precursors could be catalyzed by microorganisms or enzymes to form new compounds or derivatives. Moreover, some functional-group modifications of biotransformed derivatives might increase the bioactivity of the precursor compounds. Therefore, enzymatic biotransformation supports an efficient method of creating new and bioactive compounds used in the development of new drugs.

Biotransformation reactions include, e.g., hydroxylation, dehydrogenation, lactone formation, methylation, and (de)glycosylation [1,2,3], with the former reaction attracting greater scientific attention. The hydroxyl group-modified precursor compound may increase the bioactivities of the original precursor compounds and sometimes gain novel bioactivity after modifying the original precursors [4]. For example, tyrosinase (TYR) is a key enzyme in melanogenesis, where TYR catalyzes the oxidation of l-tyrosine to form dopaquinone and then to form melanin [5]. Recently, *Bacillus megatherium* tyrosinase (*Bm*TYR) successfully biotransformed the flavonoids precursors to produce 3′-hydroxylation derivatives [6,7].

Generally, the precursors used in biotransformation might be pure compounds or crude extracts. Using pure compounds to study biotransformation is a well-established method; however, the high cost of pure precursors limits the development of the experimental research. In contrast, commercial herb extracts offer vast amounts of diverse precursors at a low cost. Thus, we are interested in discovering new bioactive compounds from herb extracts. Recently, a biotransformation-guided purification (BGP) with *Bm*TYR was applied to the Chinese herb licorice extract, and the bioactive compound butin was produced [8]. The case study implied that the BGP process could be applied to other herbs. As described above, using herb crude extracts as precursors in the BGP process has greater advantages than using pure compounds with enzymatic biotransformation.

Ha-Soo-Oh (or He-Shou-Wu) in China is a popular traditional Chinese herb medicine that has been widely used in Asian countries as a tonic agent and nutritional supplement for centuries [9]. The herb is prepared from the dried roots of *Polygonum multiflorum* Thunb. More than 103 constituents have been isolated from the roots of *P. multiflorum* and identified as stilbene glucosides, anthraquinones, flavonoids, and other compounds [10]. Modern pharmacological studies and clinical practices have indicated that these compounds have various biological activities, including anti-tumor, anti-oxidative, anti-bacterial, anti- hyperlipidemia, anti-atherosclerosis, immunomodulating, and hepatoprotective effects [11]. Therefore, the BGP process with *Bm*TYR was applied to commercial Ha-Soo-Oh for biotransformed compounds from various precursors in the herb. Both the biotransformed compound and the possible precursor in the herb were identified, and the anti-oxidative activity of both compounds was evaluated and compared.

## 2. Results and Discussion

### 2.1. Biotransformation of Ha-Soo-Oh Extracts by BmTYR

To discover new bioactive compounds from natural extracts, four brands of commercial Ha-Soo-Oh herb medicine, including Sun Ten, Ko Da, Chuang Song Zong, and Min Tong, were selected as the raw materials. The methanol extracts of the four Ha-Soo-Oh herbs were analyzed with high-performance liquid chromatography (HPLC), and the result showed that three of them, that is, Sun Ten, Ko Da, and Chuang Song Zong, contained one major constituent (Figure 1). In contrast, the crude extract of Min Tong Ha-Soo-Oh contained more complex constituents. Accordingly, the crude extract of Sun Ten Ha-Soo-Oh was selected due to its high content of the major constituent. The efficacy of biotransformation by *Bm*TYR toward the extract was evaluated. The HPLC analysis showed that a biotransformed product, compound (**1**), can be directly catalyzed from the major constituent in the extract by *Bm*TYR (Figure 2).

### 2.2. Purification and Identification of the Biotransformation Product from the Ha-Soo-Oh Extract

To resolve the chemical structure of compound (**1**), the biotransformation was scaled up to 50 mL. The biotransformation product was purified by preparative HPLC. The chemical structure of purified compound (**1**) was then analyzed using high-resolution mass spectrometry (HRMS) and nucleic magnetic resonance (NMR) spectral analyses. The molecular formula of compound (**1**) was established as C_20_H_22_O_10_ by the HRMS at *m*/*z* 421.1129 [M-H]^−^ (Appendix A). In addition, the ^1^H-NMR spectra (700 MHz, DMSO-*d_6_*) provided the following chemical shits *δ*_H_: 3.22 (1H, m, H-5″), 3.26 (1H, m, H-4″), 3.28 (1H, m, H-3″), 3.36 (1H, m, H-2″), 3.55 (1H, m, H-6a″), 3.68 (1H, m, H-6b″), 4.41 (1H, d, *J* = 7.7 Hz, H-1″), 6.17 (1H, d, *J* = 2.8 Hz, H-4), 6.53 (1H, d, *J* = 2.8 Hz, H-6), 6.70 (1H, d, *J* = 7.7 Hz, H-5’), 6.80 (1H, d, *J* = 16.5 Hz, H-8), 6.90 (1H, dd, *J* = 2.1, 7.7 Hz, H-6’), 7.00 (1H, *J* = 2.1 Hz, H-2’), 7.61 (1H, d, *J* = 16.5 Hz, H-7). The results observed in the ^13^C-NMR spectra (175 MHz, DMSO- *d_6_*) were as follows: *δ*_C_ 60.7 (C-6″), 69.4 (C-4″), 74.0 (C-2″), 76.1 (C-3″), 77.1 (C-5″), 101.0 (C-6), 102.6 (C-4), 106.5 (C-1″), 113.8 (C-2’), 115.7 (C-5’), 118.6 (C-6’), 120.4 (C-7), 128.7 (C-1’), 129.1 (C-1), 131.9 (C-8), 136.3 (C-2), 145.2 (C-3’), 145.4 (C-4’), 150.5 (C-3), 154.6 (C-5). The NMR spectra exhibited characteristic glucosyl signals: the anomeric carbon signal at δ 106.5, one CH_2_ signal at *δ* 60.7; and four CH signals at *δ* 77.1, 76.1, 74.0, 69.4. The large coupling constant (7.7 Hz) of the anomeric proton H-1″ (*δ* 4.41 ppm) indicated the *β*-configuration. In the aromatic protons spectrum, two protons as an AX pattern at 6.17 (d, *J* = 2.8 Hz) and 6.53 (d, *J* = 2.8 Hz), assigned to H-4 and H-6, respectively, with the remaining three of the aromatic protons appeared as an A’B’X’ pattern at 7.00 (d, *J* = 2.1 Hz), 6.70 (d, *J* = 7.7 Hz), and 6.90 (dd, *J* = 7.7, 2.1 Hz) assigned to H-2’, H-5’ and H-6’, respectively. The double bond protons H-7 (d, *J* = 16.5 Hz) and H-8 (d, *J* = 16.5 Hz) were obtained from the homonuclear correlation spectroscopy (COSY) spectrum. The full assignments of the ^1^H- and ^13^C-NMR signals were further aided by distortionless enhancement by polarization transfer (DEPT), heteronuclear single quantum coherence (HSQC), heteronuclear multiple-bond connectivity (HMBC), COSY, and nuclear Overhauser effect spectroscopy (NOESY) spectra, as shown in Appendix A. The cross peak of glucosyl H-1″ with C-2 (4.41/136.3 ppm) in the HMBC spectrum demonstrated the structure of compound (**1**) to be PSG (Figure 3). There are many stilbene glycosides identified from the roots of the plant *P. multiflorum* [12,13,14]; however, PSG is newly discovered and has never been isolated from the plant. Moreover, to our knowledge, PSG is a new compound.

### 2.3. Confirming the Biotransformation Process

The HPLC analysis revealed that PSG was the biotransformation product of the main constituent in the Ha-Soo-Oh extract (Figure 2). In addition, the structure of PSG implied that it was the hydroxylation product of 3,5,4′-trihydroxystilbene-2-*O*-*β*-glucoside (TSG), which is the main stilbene glucoside in the roots of *P. multiflorum* [15,16,17] and considered the quality index of the Ha-Soo-Oh herb medicine in the 2015 edition of the Chinese pharmacopoeia. Thus, it was hypothesized that the produced PSG was from the hydroxylation of TSG by *Bm*TYR. To confirm the hypothesis, the structure of commercial TSG standard was confirmed by HRMS (Appendix A), and the TSG precursor was biotransformed by *Bm*TYR into new products for HPLC analysis. The result clearly showed that the retention time of TSG in the HPLC analysis was identical to that of the major constituent in the crude extract of Sun Ten Ha-Soo-Oh (Figure 4a). Moreover, the retention time of the biotransformed product was identical to that of the compound (**1**), PSG (Figure 4b). Hence, the biotransformed product PSG from the Ha-Soo-Oh extract was from the precursor TSG in Ha-Soo-Oh. Figure 5 illustrates the biotransformation process of TSG by *Bm*TYR.

Recently, some research has focused on the biotransformation of stilbenes or stilbene glycosides. For examples, Shimoda et al. used *Phytolacca americana*, the plants cells to biotransform two natural polyhydroxystilbenes, which have pharmacological activities. They found that the plant cells may convert oxyresveratrol (3,5,2′,4′-tetrahydroxystilbene) and gnetol (3,5,2′,6′-tetrahydroxystilbene) to 3-, 2′-, and 4′-*β*-glucosides of oxyresveratrol and 2′-*β*-glucoside of gnetol, respectively [18]. Lin et al. used *β*-glucosidase of a medicinal fungus *Ganoderma lucidum* to deglycosylate the TSG into TSG aglycone (2,3,5,4′-tetrahydroxystilbene) [19]. This study further revealed *Bm*TYR could glycosylate TSG of Ha-Soo-Oh into the new stilbene glycoside (PSG) via a BGP process. The BGP process is economic and easier to scale-up for industrial applications.

### 2.4. Antioxidant Activity of TSG and PSG

It has been reported that the *ortho*-dihydroxyl groups on the benzene ring of flavonoid structures play a vital role in exerting its antioxidant activity [20]. Thus, the anti-oxidative activities of both PSG and its precursor TSG were determined by the 1,1-Diphenyl-2-picrylhydrazine (DPPH) free radical-scavenging assay. The assay showed that the antioxidant activity of PSG was comparable to that of ascorbic acid, which was 4.7-fold higher than that of TSG (Figure 6).

The Chinese herb, Ha-Soo-Oh, has been used to prevent canities (hair graying) and skin hyperpigmentation for a long period [9,21]. Modern scientific research has demonstrated that TSG is the main bioactive constituent with the melanogenesis stimulating activity [22,23]. In addition, TSG has broad biological actions, including anti-aging effects, prevention, and treatment of Alzheimer’s disease (AD) and Parkinson’s disease (PD), anti-oxidation scavenging of free radicals, lowering of cholesterol levels, anti-atherosclerosis effects, anti-inflammatory effects, liver protection, diastolic blood vessels, and anti-tumor effects [15,16,17]. PSG is a TSG analog with functional *ortho*-dihydroxyl groups on the benzene ring, which greatly improve its antioxidant activity (Figure 6). Thus, PSG might also possess multiple bioactivities and the newly identified stilbene glucoside is a highly potential analog for the development of new drugs.

## 3. Materials and Methods

### 3.1. Enzymes and Chemicals

Four different brands of commercial Ha-Soo-Oh herb medicine, including Sun Ten, Ko Da, Chuang Song Zong, and Min Tong, were purchased from a local medicine store, and the TSG standard was purchased from Baoji Herbest Bio-Tech (Xi-An, Shaanxi, China). DPPH, dimethyl sulfoxide (DMSO), and ascorbic acid were purchased from Sigma (St. Louis, MO, USA). Recombinant *Bm*TYR with a specific l-dihydroxyphenylalanine (l-DOPA) oxidation activity of 4.84 U/mg was prepared from the previous study [8], and the other reagents and solvents used were commercially available.

### 3.2. Preparation of Ha-Soo-Oh Extract

The commercial Ha-Soo-Oh powder (10 g) was extracted with 100 mL of 50% of methanol at 25 °C for 24 h. The mixture was filtrated with Watman filter paper, and the filtrate was condensed under a vacuum, and then dehydrated by freeze drying. The condensed mixture was dried by freeze drying. Finally, 0.23, 0.26, 0.18, and 0.34 g of the extracts were obtained from Sun Ten, Ko Da, Chuang Song Zong, and Min Tong Ha-Soo-Oh, respectively.

### 3.3. Biotransformation Using BmTYR

The biotransformation system was operated according to the previous report, with minor modifications [8]. The reaction mixture (100 μL), containing 500 mM of borate (pH 9.0), 10 mM of ascorbic acid, and 2 mg/mL of the crude extract or 1 mg/mL of TSG (diluted from a stock of 20 mg/mL in DMSO) and 120 μg/mL of *Bm*TYR, was incubated at 50 °C and shaken at 200 rpm for 1.5 h. At the end of the reaction, 20 μL of 1 M HCl and 120 μL of MeOH were added to stop the reaction, which was analyzed using HPLC.

### 3.4. HPLC

A combo system was used that consisted of an Agilent^®^ 1100 series HPLC system (Agilent, Santa Clara, CA, USA) equipped with a gradient pump (Waters 600, Waters, Milford, MA, USA). The HPLC system was controlled via a PC workstation using Chromatography Data Station software (SISC, Scientific Information Service Co., LTD., New Taipei, Taiwan). The stationary phase was a C18 column (5 μm, 4.6 i.d. × 250 mm; Sharpsil H-C18, Sharpsil, Beijing, China), and the mobile phase was 1% acetic acid in water (A) and methanol (B). The elution condition was a linear gradient from 0 min with 40% B to 20 min with 70% B, an isocratic elution from 20 min to 25 min with 70% B, a linear gradient from 25 min with 70% B to 28 min with 40% B, and an isocratic elution from 28 min to 35 min with 40% B. All eluants were eluted at a flow rate of 1 mL/min. The sample volume was 10 μL, and the detection condition was set at 254 nm.

### 3.5. Purification and Identification of the Biotransformation Metabolite

The purification process was a previously described method [8]. To purify compound (**1**), the biotransformation reaction was scaled up to 50 mL containing 100 mg of the extract of Sun Ten Ha-Soo-Oh, which was equally divided into 100 1.5 mL tubes, and the reaction was conducted in an incubator at 50 °C and with 200 rpm shaking for 1.5 h. After the reaction, compound (**1**) was purified by a preparative YoungLin HPLC system (YL9100, YL Instrument, Gyeonggi-do, South Korea). The elution corresponding to the peak of compound (**1**) in the HPLC analysis was collected, concentrated by a rotary evaporator (Rotavapor^®^ R-100, Bunchi company, Tokyo, Japan), and then dehydrated by a freeze dryer (Kingmech Sci. Co., LTD., New Taipei, Taiwan). Finally, 20.3 mg of compound (**1**) was obtained. The yield of compound (**1**) is 20.3% from the crude extract of Ha-Soo-Oh. The structure of the compound was confirmed with nucleic magnetic resonance (NMR) and mass spectral analyses. The mass analysis was performed using a HRMS spectrometer (Q-Exactive Plus, Thermo Fisher Scientific, Waltham, MA, USA). ^1^H- and ^13^C-NMR, DEPT, HSQC, HMBC, COSY, and NOESY spectra were recorded on a Bruker AV-700 NMR spectrometer (Bruker Company, Billerica, MA, USA) at ambient temperature. Standard pulse sequences and parameters were used for the NMR experiments, and all chemical shifts were reported in parts per million (ppm, *δ*).

### 3.6. Determination of DPPH Free Radical-Scavenging Activity

The assay was performed as previously described [8], with minor modifications. The tested sample (dissolved in DMSO) was added to the DPPH solution (1 mM in methanol) to a final volume of 0.1 mL. After 15 min of the reaction, the absorbance of the reaction mixture was measured at 517 nm with a microplate reader (Sunrise, Tecan, Männedorf, Switzerland). Ascorbic acid (dissolved in DMSO) was used as a positive antioxidant standard. The DPPH free radical-scavenging activity was calculated as follows:DPPH free radical scavenging-activity = (OD_517_ of the control reaction − OD_517_ of the reaction)/(OD_517_ of the control reaction).(1)

The concentration of an inhibitor required to scavenge 50% of the initial DPPH free radicals under the assay conditions is defined as the IC_50_ value.

## 4. Conclusions

A new stilbene glucoside, PSG, was catalyzed by *Bm*TYR from crude extract of a Chinese herb (Ha-Soo-Oh), which is directly isolated via the biotransformation-guided purification (BGP) process. The structure of PSG further revealed the precursor TSG isolated from Ha-Soo-Oh could be catalyzed by *Bm*TYR. The newly identified PSG possessed comparable DPPH free radical-scavenging activity with that of ascorbic acid, which was 4.7-fold higher than that of its precursor, TSG. The present study showed that the BGP is an alternative process to obtain new bioactive molecules directly from herb crude extracts.

## Figures and Tables

**Figure 1 plants-11-02286-f001:**
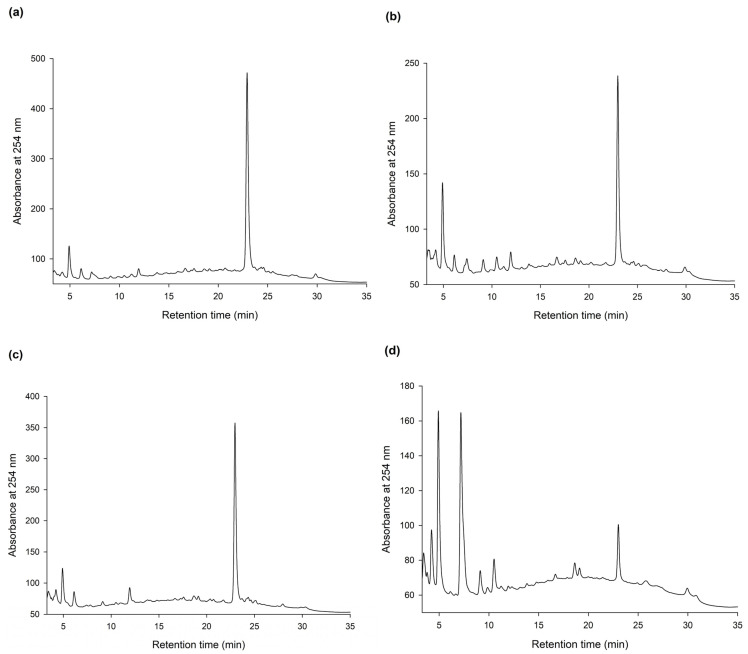
High-performance liquid chromatography (HPLC) analysis of the methanol extracts (5 mg/mL) from four different brands, Sun Ten (**a**), Ko Da (**b**), Chuang Song Zong (**c**), and Min Tong (**d**), of Ha-Soo-Oh herb medicine. The HPLC operation procedure is described in Section 2.4.

**Figure 2 plants-11-02286-f002:**
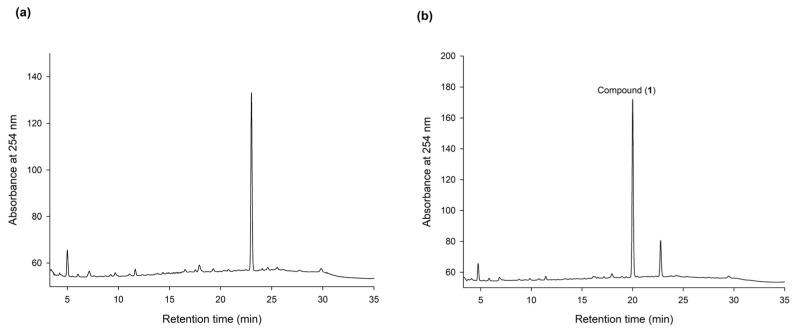
HPLC analysis of the methanol extract of Sun Ten Ha-Soo-Oh herb medicine (**a**) and the biotransformed products by using *Bm*TYR (**b**). The biotransformation mixture—containing 120 μg/mL of the purified recombinant *Bm*TYR enzymes, 2 mg/mL of the extract, 10 mM of ascorbic acid, and 500 mM of borate buffer at pH 9—was incubated at 50 °C and shaken at 200 rpm for 1.5 h. At the end of the reaction, one-fifth of the volume of 1 M HCl and an equal volume of MeOH were added to stop the reaction, and it was analyzed using HPLC. The HPLC operation procedure is described in Section 2.4. The retention times of the major peak in (**a**) and compound (**1**) in (**b**) are 23.2 and 20.5 min, respectively.

**Figure 3 plants-11-02286-f003:**
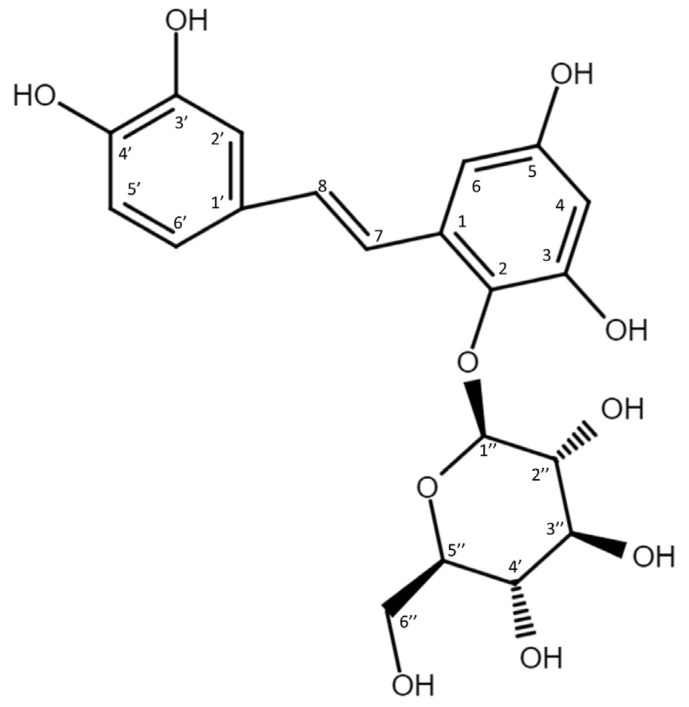
Chemical structure of the biotransformation product (**1**), 2,3,5,3′,4′-pentahydroxystilbene-2-*O*-*β*-glucoside (PSG).

**Figure 4 plants-11-02286-f004:**
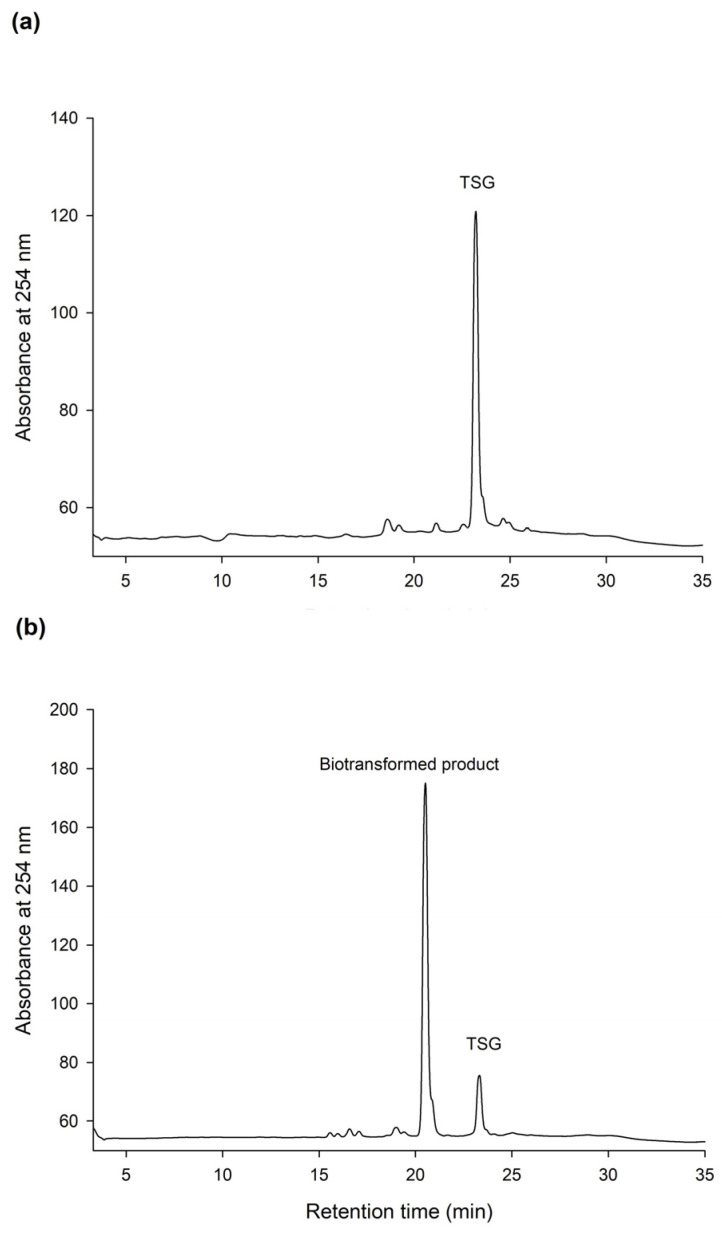
HPLC analysis of 3,5,4′-trihydroxystilbene-2-*O*-*β*-glucoside (TSG) (**a**) and the biotransformed product by using *Bm*TYR (**b**). The biotransformation and HPLC conditions were the same as those in the legend of Figure 1, with the replacement of the enzyme substrate (2 mg/mL of Ha-Soo-Oh crude extract) with a TSG standard (2 mg/mL). The retention times of TSG in and the biotransformed product are 22.3 and 20.5 min, respectively.

**Figure 5 plants-11-02286-f005:**
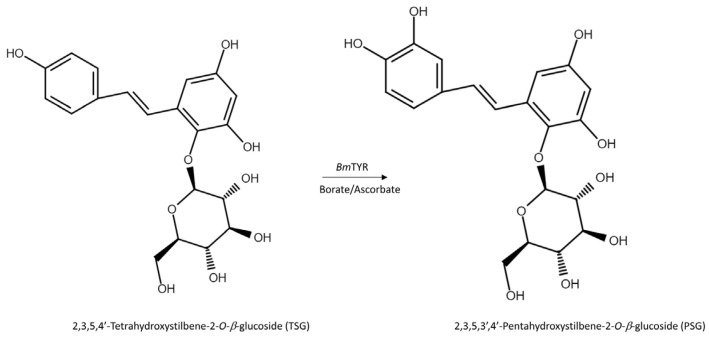
The biotransformation process of TSG by *Bm*TYR.

**Figure 6 plants-11-02286-f006:**
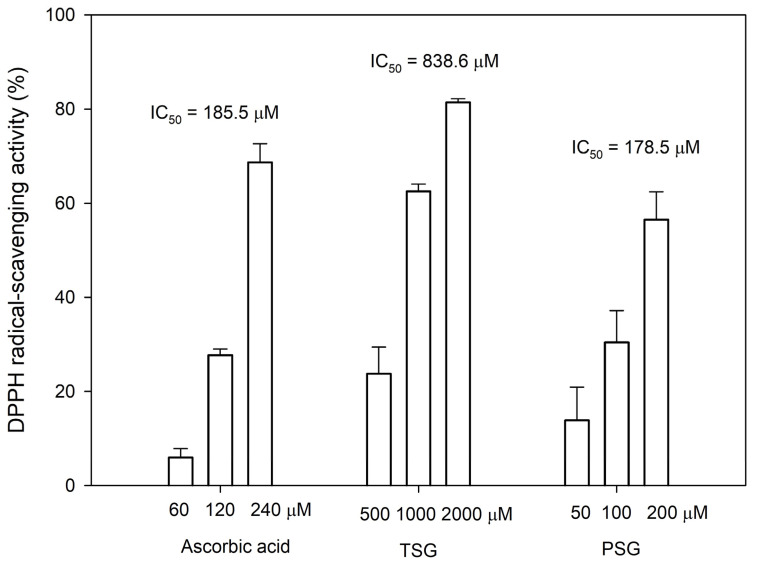
1,1-Diphenyl-2-picrylhydrazine (DPPH) free radical-scavenging activity of TSG, PSG, and ascorbic acid. The DPPH scavenging activity was determined as described in Materials and Methods. The mean (*n* = 3) is shown, and the standard deviation (S.D.) is represented by error bars. The IC_50_ values represent the concentrations required for 50% DPPH free radical-scavenging activity.

## Data Availability

Not applicable.

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
