# Peer review of "A New Stilbene Glucoside from Biotransformation-Guided Purification of Chinese Herb Ha-Soo-Oh"

_plants, 2022, doi:10.3390/plants11172286_

Round 1
Reviewer 1 Report
This manuscript uses Polygonum multiflorum as a starting material to obtain the highest yields on hydroxylated phenolics with antioxidant potential.
In general, the manuscript is well redacted. However, it lacks literature references to support mainly what is being argued (in all the manuscript). In fact, it totals 15 references. This is a poor discussion as the authors are reporting new molecular structures.
Conclusions lines 230-232: this section needs an improvement. I would say that your work is an alternative and a stepping stone moved for the synthesis of new molecules with bioactive potential by the pharmaceutical industry.
Author Response
Dear Reviewer,
Thank you very much for your helpful comments. We have revised according to the comments point-to-point. All responses are in the attached file. Please check it. Thank you again for your help to improve the manuscript.

Reviewer 2 Report
Interesting article about the biotransformation (hydroxylation) of an already known stilbene derivative isolated from plant material.
General remarks:
The chromatograms shown in the figures do not have a clear scale. Are these normalized to 100% or is a Y-axis with absolute values used? in the last case, please indicate each time the corresponding scale (a base line cannot start at 800 nm!) otherwise the comparison is not possible. Furthermore, please annotate the known peak (e.g. TSG). Finally, it would improve the understanding if the mass of this precursor would be shown.
The determination of a chemical form is not possible with low-resolution MS instruments. Considering the mass accuracy of the MS instrument used (421.3±0.1) and the elements C, H, N, and O only (no S for example), more than 200 chemical formulae have to be considered for the structure elucidation. Therefore, the statement in L97 'the molecular formula of 1 was established as C20H22O10 by ESI-MS' is wrong!
A table containing the assignement of the NMR data of compound 1 is lacking. As an example, see table 1 in Helv. Chim. Acta, 2005, 88, p23 ( DOI 10.1002/hlca.200490293). Also signals corresponding to contamination have to be interpreted (e.g. below 3 ppm in the 1H-NMR)
The HPLC retention time as a physical property should be mentioned, in particular when compounds from two samples are compared (fig. 4)
I have also the following short remarks:
L38: Biotransformation reactions include, e.g., hydroxylation...
L166-168: Wording is wrong (bioactivates?)
L193-195: an Agilent 1100 system cannot be equipped with a Waters gradient pump. There are not compatible.
L208: a solution cannot be condensed under a vacuum...
Author Response

(The authors gave the same response as above.)

Reviewer 3 Report
This manuscript reported a new stilbene glucoside PSG transformed from the main component TSG in Chinese herb medicine Ha-Soo-Oh through the BGP process with BmTYR as a biocatalyst, including biotransformation, preparative HPLC isolation, and structure identification. The new compound was also found to have potent DPPH free radical scavenging activity. It is a meaningful research for development and potential utilization of BGP in herbs. However, there are some problems in the MS, and below is the major:
1. Compared with TSG, PSG is only a hydroxylation compound. Therefore, the word “new” is a suitable adjective rather than “novel”.
2. As a new compound, HRMS is indispensable. However, in this paper, only ESI-MS was found, please replenish.
3. In identification section, most of the 1H- and 13C- NMR data were repeated twice. In preparation section, the condensed mixture was dehydrated by freeze drying, after that, it was then dried by freeze drying again. It is difficult to understand freeze drying two times. Please check.
4. In purrification and identification section: what is the material? Extract or TSG? The author described that 20.3 mg of cpd 1 was obtained, what is the transformation yield? Please clarify these key problems.
5. In HPLC section, Agilent 1100 series HPLC system is the equipment, why is it equipped a Waters 600 pump?
6. Atom number of the structure of PSG should be provided for better understanding the NMR data.
7. Line 112, H4 and H6 should be H-4 and H-6, line 116, 1H and 13C-NMR should be 1H- and 13C-NMR.
8. For the legneds in Supplemental Materials, 1D NMR, 2D NMR… should be replaced with 1H NMR, 13C NMR, and COSY, NOESY, HMBC… etc..
In conclusion, I think if the author can solve the above problems, publication should be considered.
Author Response

(The authors gave the same response as above.)

Round 2
Reviewer 1 Report
The authors greatly addressed the reviewer's comments improving the quality of the presented work. Therefore, I don't have further answers.